# Structures of translationally inactive mammalian ribosomes

Alan Brown[1]*, Matthew R Baird[1], Matthew CJ Yip[2], Jason Murray[3], Sichen Shao[2]*

[1]Department of Biological Chemistry and Molecular Pharmacology, Harvard Medical School, Boston, United States; [2]Department of Cell Biology, Harvard Medical School, Boston, United States; [3]MRC Laboratory of Molecular Biology, Cambridge, United Kingdom

**Abstract** The cellular levels and activities of ribosomes directly regulate gene expression during numerous physiological processes. The mechanisms that globally repress translation are incompletely understood. Here, we use electron cryomicroscopy to analyze inactive ribosomes isolated from mammalian reticulocytes, the penultimate stage of red blood cell differentiation. We identify two types of ribosomes that are translationally repressed by protein interactions. The first comprises ribosomes sequestered with elongation factor 2 (eEF2) by SERPINE mRNA binding protein 1 (SERBP1) occupying the ribosomal mRNA entrance channel. The second type are translationally repressed by a novel ribosome-binding protein, interferon-related developmental regulator 2 (IFRD2), which spans the P and E sites and inserts a C-terminal helix into the mRNA exit channel to preclude translation. IFRD2 binds ribosomes with a tRNA occupying a noncanonical binding site, the 'Z site', on the ribosome. These structures provide functional insights into how ribosomal interactions may suppress translation to regulate gene expression.

DOI: https://doi.org/10.7554/eLife.40486.001

*For correspondence:
alan_brown@hms.harvard.edu (AB);
sichen_shao@hms.harvard.edu (SS)

Competing interests: The authors declare that no competing interests exist.

## Introduction

Translation is an important point of regulation for gene expression. The overall levels and activities of ribosomes are implicated in cellular differentiation, developmental disorders, and cancers (*Buszczak et al., 2014*; *Narla and Ebert, 2010*). Red blood cell differentiation and function is especially sensitive to ribosome concentrations and activity. Humans produce over two million red blood cells every second (*Dzierzak and Philipsen, 2013*), with hemoglobin comprising 98% of total protein content of each cell. To achieve this, translation is optimized for hemoglobin production during red blood cell differentiation (*Mills et al., 2016*; *Smith and McNamara, 1971*). This translational program occurs predominantly in enucleated cells, precluding transcriptional control and relying solely on preexisting ribosomes and translational factors before the ribosomes are degraded upon terminal differentiation (*Rowley, 1965*).

Underscoring the importance of ribosomal activity in blood cell differentiation, mutations in ribosomal proteins, ribosome-assembly factors, and the ribosome-degradation machinery manifest as blood disorders (*Draptchinskaia et al., 1999*; *Ebert et al., 2008*) and are often found in cancers of blood cells (*De Keersmaecker et al., 2013*; *Ljungström et al., 2016*). A recent study demonstrated that mutations that cause Diamond-Blackfan anemia alter global ribosome levels, which dictates the mRNAs that are translated (*Khajuria et al., 2018*). Similarly, mutations in the ubiquitination machinery required to eliminate ribosomes during terminal red blood cell differentiation results in anemia in mice (*Nguyen et al., 2017*).

Although controlling absolute ribosome levels is an effective mechanism to regulate gene expression, assembling and degrading ribosomes is slow and requires significant energy consumption (*Kressler et al., 2010*; *Warner, 1999*). For faster responses, two mechanisms are known to acutely

and reversibly repress global translation in eukaryotes. The first is phosphorylation of translation initiation factor eIF2α, which dampens protein synthesis by preventing the formation of productive translation initiation complexes (*Wek et al., 2006*). In mammals, four distinct kinases phosphorylate eIF2α in response to different cellular stresses, including nutrient deprivation, endoplasmic reticulum (ER) stress, viral infections, and heme deprivation (*Chen, 2014*). The second mechanism is translational repression by the yeast Stm1 protein or its mammalian homolog, SERPINE mRNA-binding protein 1 (SERBP). Stm1 directly binds and sequesters 80S ribosomes (*Balagopal and Parker, 2011*; *Van Dyke et al., 2006*). This function of Stm1 is thought to preserve ribosomes during nutrient deprivation (*Van Dyke et al., 2013*) and may influence the degradation of some mRNAs (*Balagopal and Parker, 2011*).

Despite emerging evidence of the importance of global translation levels on short- and long-term gene regulation, other mechanisms for modulating ribosome activities remain poorly characterized. Here, we use electron cryomicroscopy (cryo-EM) to analyze translationally inactive ribosomes isolated from mammalian reticulocytes, the penultimate stage of red blood cell differentiation (*Dzierzak and Philipsen, 2013*). In addition to observing ribosomes silenced by SERBP1, we identify interferon-related developmental regulator 2 (IFRD2) as a novel factor capable of translationally inactivating ribosomes. IFRD2 directly binds 80S ribosomes, precluding binding of mRNA and P-site tRNA. We additionally observe deacylated tRNA occupying a noncanonical binding site on the mammalian ribosome that is exploited by viral internal ribosome entry site (IRES)-mediated translation. We observe tRNA in this site in the presence and absence of either IFRD2 and P-site tRNA. Together, these findings identify new ribosomal interactions that may modulate global translation activity during erythropoiesis and in other differentiating cells.

## Results

### Identification of translationally inactive ribosomes in reticulocyte lysate

We previously used a cell-free translation system derived from rabbit reticulocyte lysate to isolate various ribosomal complexes for cryo-EM analysis (*Brown et al., 2015b*; *Shao et al., 2016*). Despite biochemical enrichment for specific complexes, the target structures generally derived from 10% to 20% of the particles within each cryo-EM dataset. The remaining particles are various translation intermediates and inactive ribosomes that copurify with the target complex. To characterize these reticulocyte ribosomal populations, we reanalyzed a combined cryo-EM dataset of 971,191 ribosomal particles purified from in vitro translation reactions containing a dominant-negative release factor (DN-eRF1) added to enrich for translational termination complexes (*Brown et al., 2015b*). These datasets were imaged under identical conditions, with the only difference being the identity of the stop codon in the mRNA used to program the in vitro translation reactions. As all stop codons adopt the same structural fold (*Brown et al., 2015b*), these samples can be considered biochemically and structurally analogous.

Three-dimensional classification of the combined dataset of ribosomal particles identified nine structurally distinct classes of 80S ribosomes (*Figure 1*; *Table 1*). These included the target termination complex and known intermediates of translation. Notably, we did not observe any empty 80S ribosomes lacking a bound tRNA or protein factor. Four classes represented non-translating ribosomes based on the absence of canonical tRNAs. These structures fall into three types of translationally inactive 80S ribosomes (1) bound to eEF2 and SERBP1, (2) bound to IFRD2, and (3) containing a tRNA in a noncanonical binding site.

### SERBP1 traps eEF2 on different conformations of inactive ribosomes

The two most abundant classes of non-translating ribosomes are distinct conformations (rotated and unrotated) of 80S ribosomes bound to the elongation factor eEF2 (*Figure 2—figure supplement 1A*). eEF2 normally catalyzes the translocation of mRNA and peptidyl-tRNA during translation, although its association with inactive ribosomes is well established (*Liu and Qian, 2016*). Closer inspection reveals that all maps of non-translating ribosomes with eEF2 in our dataset also contain SERBP1 in the mRNA entrance channel (*Figure 2A,B*). It is possible that eEF2 requires SERBP1 to form stable interactions with inactive ribosomes. Our map resolution (3.4 Å) allows the reassignment of SERBP1 residues from a 5.4 Å-resolution cryo-EM structure of eEF2-SERBP1 on the human

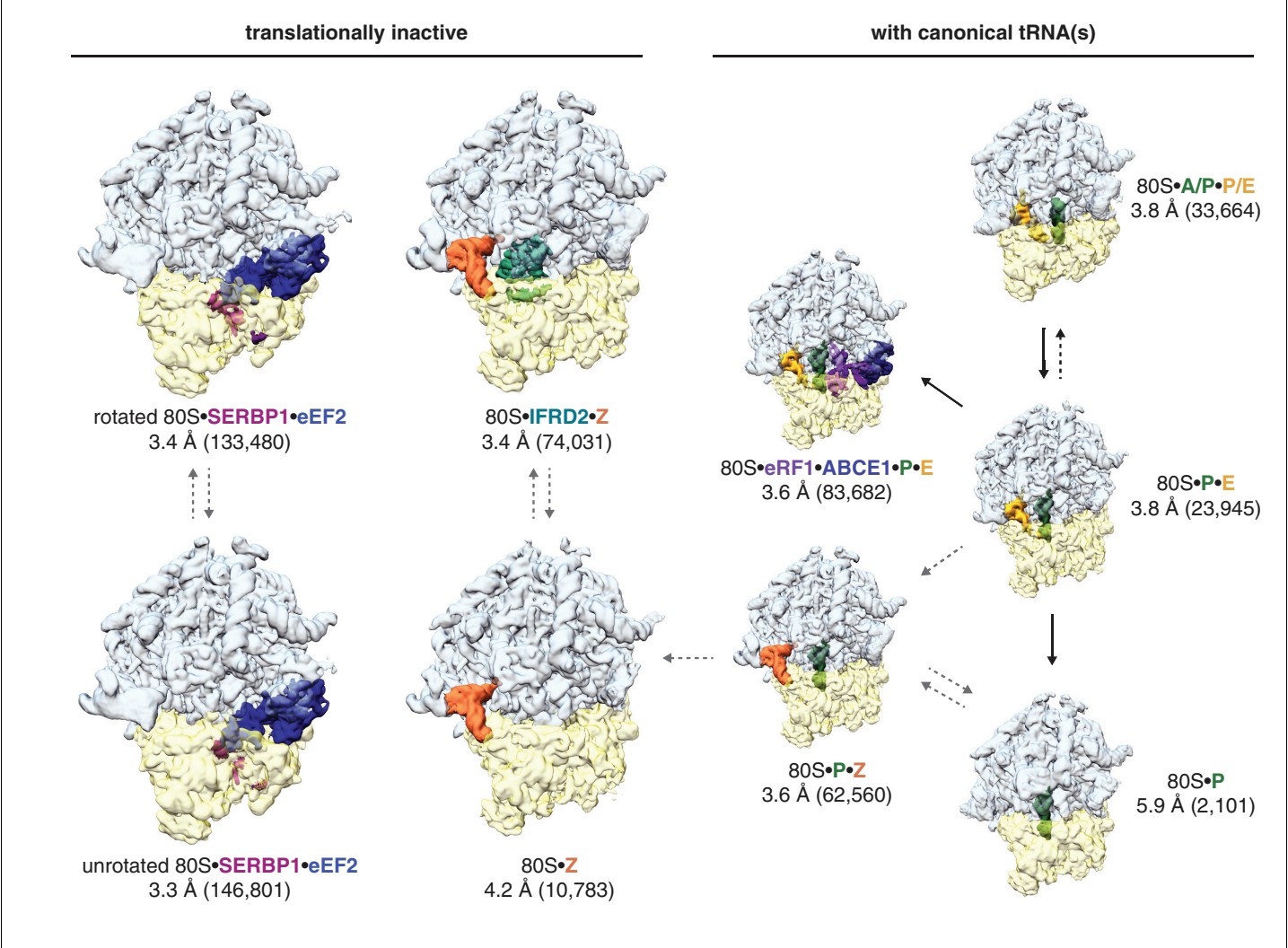

**Figure 1.** Identification of translationally inactive ribosomal complexes in a cryo-EM dataset. In silico classification of a cryo-EM dataset of ribosomal particles reveals known intermediates of translation and translationally inactive 80S ribosomes. The overall resolution and number of particles that make up the reconstruction are shown for each complex.

DOI: https://doi.org/10.7554/eLife.40486.002

The following figure supplement is available for figure 1:

**Figure supplement 1.** Global and local resolution.

DOI: https://doi.org/10.7554/eLife.40486.003

ribosome (*Anger et al., 2013*) to produce a model that is more consistent with the structure of Stm1 on the yeast ribosome (*Ben-Shem et al., 2011*) and cross-linking data (*Liu et al., 2015*).

SERBP1 and Stm1 interact with the A and P sites of the ribosomal mRNA channel via conserved residues (*Figure 2—figure supplement 2*), including a DRHS motif (residues 194–197 in SERBP1) that forms a $3_{10}$-helix around nucleotide C1701 of 18S rRNA (*Figure 2C,D*; *Figure 2—figure supplement 1B*). The density of SERBP1 is less clear following residue 201 in the unrotated ribosome (*Figure 2—figure supplement 1C*), potentially the result of averaging splice variants that differ at this site (*Figure 2—figure supplement 2*). The C-termini of SERBP1 and Stm1 emerge from the mRNA channel to interact with eS10, eS12 and eS31, although the density is only sufficiently resolved to model a short helix of SERBP1 (*Figure 2—figure supplement 1D*). Unlike Stm1, we observe no interaction between the N terminus of SERBP1 and the ribosomal large subunit.

At the A site, SERBP1 residues 198 – 201 interact with domain IV of eEF2 (*Figure 2E*, *Figure 2—figure supplement 1E*). This small interface is unlikely to physically anchor eEF2 to the ribosome,

**Table 1.** Structurally distinct classes identified within a cryo-EM dataset of 971,191 ribosomal particles.

| Ribosomal complex | Resolution (Å) | Particles | % of particles | EMDB accession code |
|---|---|---|---|---|
| 80S classes (unrotated) | | | | |
| 80S • P | 5.9 | 2101 | 0.2 | 9234 |
| 80S • P • E | 3.8 | 23,945 | 2.5 | 9235 |
| 80S • P • E • eRF1 • ABCE1 | 3.6 | 83,682 | 8.6 | - |
| 80S • Z | 4.2 | 10,783 | 1.1 | 9236 |
| 80S • Z • P | 3.6 | 62,560 | 6.4 | 9237 |
| 80S • Z • IFRD2 | 3.4 | 74,031 | 7.6 | 9239 |
| 80S • eEF2 • SERBP1 (head swivel) | 3.3 | 146,801 | 15.1 | 9240 |
| 80S classes (rotated) | | | | |
| 80S • A/P • P/E | 3.8 | 33,664 | 3.5 | 9241 |
| 80S • eEF2 • SERBP1 | 3.4 | 133,480 | 13.7 | 9242 |

DOI: https://doi.org/10.7554/eLife.40486.011

and the presence of GDP in the nucleotide-binding pocket of eEF2 (*Figure 2—figure supplement 1F*) despite differences in the switch-loop conformations (*Figure 2—figure supplement 1G*) indicates that SERBP1 does not impair GTP hydrolysis by eEF2. The exact mechanism for how SERBP1 is able to trap eEF2 on the ribosome is unclear, although there are similarities with the eEF2-inhibitor sordarin (*Pellegrino et al., 2018*). Like SERBP1, sordarin traps eEF2 on the yeast ribosome without impairing GTP hydrolysis or inducing large-scale structural changes in eEF2 (*Pellegrino et al., 2018*). Instead, sordarin binds in a pocket between domains III, IV, and V, and increases interdomain contacts to subtly constrain the domains in conformations that prevent release. SERBP1 may also constrain domain rearrangements necessary for release through its interaction with eEF2 domain IV. Thus, SERBP1 simultaneously removes a ribosome and an abundant translational GTPase from active translation.

## IFRD2 binds the ribosomal core and is incompatible with translation

The second most abundant subset of translationally inactive 80S ribosomes contain IFRD2 in the intersubunit space of a nonrotated ribosome, where it occludes tRNA binding to the P and E sites (*Figure 3A*) and mRNA binding to the mRNA channel. We identified IFRD2 by comparing the predominantly α-helical density with the secondary structure profiles of approximately 150 candidates identified by mass spectrometry of the cryo-EM sample (*Supplementary file 1*). IFRD2 matches the number and length of the helices present in the map and has a sequence that fits regions of the map with well-defined density (*Figure 3—figure supplement 1A,B*). Supporting this assignment, both IFRD2 and its paralog IFRD1 co-immunoprecipitate with endogenously tagged ribosomes from mouse embryonic stem cells (*Simsek et al., 2017*), although IFRD1 could be excluded here based on side-chain density (*Figure 3—figure supplement 1B*). IFRD proteins are generally described as transcriptional regulators implicated in cellular differentiation and various human diseases (*Vietor et al., 2002*), although their naming as 'interferon-related' is apparently due to a mistaken sequence similarity with mouse interferon-β (*Tirone and Shooter, 1989*). Our structure suggests that IFRD proteins regulate translation instead of, or in addition to, transcription.

The core of IFRD2 (residues 71 – 342), which follows an unstructured N-terminus absent in our reconstruction, adopts an Armadillo-type fold formed by six HEAT repeats (*Figure 3B*). Each HEAT repeat contains two α-helices linked by a short loop that together form a curved α-solenoid. A small globular domain of IFRD2 (residues 343 – 400) binds the concave surface of the α-solenoid and contacts each repeat. These core domains occupy the same space as a P-site tRNA and interact with many of the same ribosomal elements (*Figure 3C*). For example, an α-helix of IFRD2 (residues 375–384) contacts H69 of 28S rRNA at the same place as the D stem of a P-site tRNA, and IFRD2 interactions with uL5 and uL13 are reminiscent of those made by the P-site tRNA acceptor arm (*Figure 3—figure supplement 1C*). IFRD2 also interacts with the ribosomal small subunit through a C-terminal

**Table 2.** Model statistics.

| | 80S · Z · P | 80S · Z · IFRD2 | 80S · eEF2 · SERBP1 (head swivel) | 80S · eEF2 · SERBP1 (rotated) |
|---|---|---|---|---|
| Model composition | | | | |
| Protein residues | 11,547 | 11,901 | 12,760 | 12,591 |
| RNA bases (+modified bases) | 5686 (40) | 5603 (26) | 5534 (36) | 5528 (40) |
| Ligands ($Zn^{2+}$/$Mg^{2+}$) | 8/297 | 8/311 | 8/297 | 8/298 |
| Refinement | | | | |
| Resolution (Å) | 3.6 | 3.4 | 3.3 | 3.4 |
| Map sharpening B factor ($Å^2$) | −101.9 | −95.5 | −97.1 | −95.1 |
| Average B factor ($Å^2$) | 75.6 | 51.8 | 84.1 | 86.1 |
| Correlation coefficient volume mask | 0.844 0.865 | 0.825 0.853 | 0.847 0.862 | 0.847 0.862 |
| Rms deviations | | | | |
| Bond lengths (Å) | 0.010 | 0.009 | 0.007 | 0.008 |
| Bond angles (°) | 1.2 | 1.1 | 1.1 | 1.0 |
| Validation (proteins) | | | | |
| MolProbity score | 1.79 | 1.64 | 1.68 | 1.71 |
| Clashscore, all atoms | 5.8 | 5.0 | 5.6 | 5.7 |
| Rotamer outliers (%) | 1.3 | 0.8 | 0.9 | 0.9 |
| EMRinger score | 2.87 | 3.24 | 2.86 | 2.97 |
| Ramachandran plot | | | | |
| Favored (%) | 94.0 | 94.3 | 94.5 | 94.0 |
| Outliers (%) | 0.1 | 0.1 | 0.1 | 0.1 |
| Validation (RNA) | | | | |
| Probably wrong sugar puckers (%) | 1.6 | 1.5 | 1.6 | 1.4 |
| Bad backbone conformations (%) | 23.6 | 23.6 | 22.8 | 22.0 |
| PDB | 6MTB | 6MTC | 6MTD | 6MTE |

DOI: https://doi.org/10.7554/eLife.40486.013

tail containing an α-helix that protrudes into the mRNA exit channel (*Figure 3D*, *Figure 3—figure supplement 1D*). Its path through the channel follows that of mRNA: it starts at the P site, extends through the E site, and ends by contacting ribosomal proteins uS11 and eS26 at the channel exit. The α-helix forms multiple electrostatic interactions with the rRNA phosphate backbone that lines the channel via highly conserved residues (*Figure 3—figure supplement 2*), suggesting that other IFRD proteins bind the ribosome with a similar insertion.

These interactions indicate that IFRD2 engages non-translating ribosomes with vacant mRNA channels and precludes further translation. Although a previous large-scale mass spectrometry study suggested that the ribosomal interaction of IFRD1 and IFRD2 are sensitive to RNase treatment (*Simsek et al., 2017*), treating reticulocyte lysate or ribosomes with RNase collapses mRNA-dependent polysomes but does not disrupt the stoichiometric association of endogenous IFRD2 with 80S ribosomes (*Figure 3E*; *Figure 3—figure supplement 3A*). This is consistent with our structure showing that IFRD2 binds to core ribosomal features independently of mRNA and tRNAs.

The observation that almost 8% of particles in the dataset contain IFRD2 suggests that it is highly abundant in reticulocytes. Indeed, IFRD2 appears to be ~8 fold less abundant on human HEK293T ribosomes compared to reticulocyte ribosomes (*Figure 3—figure supplement 3B*). Overexpressed Flag-tagged human IFRD2 in HEK293T cells also associates and copurifies with 80S ribosomes independent of RNase treatment (*Figure 3—figure supplement 3C–E*), further supporting the specific association of IFRD2 with ribosomes. We collected a small cryo-EM dataset of affinity purified

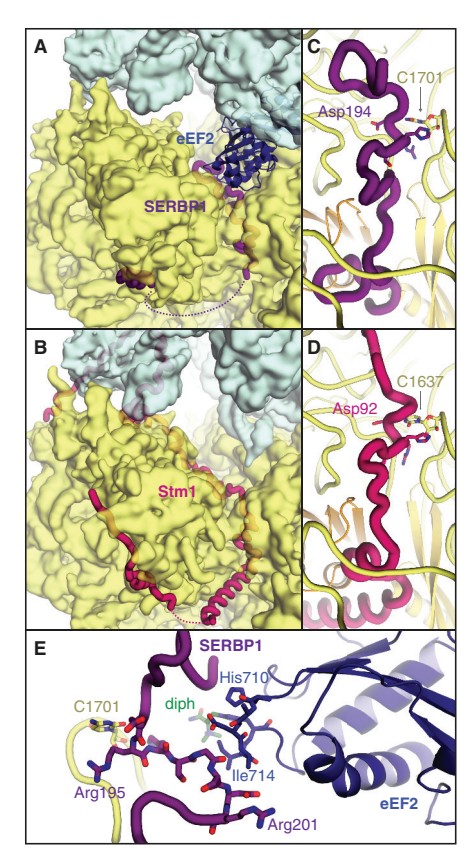

**Figure 2.** SERBP1 sequesters eEF2 on inactive ribosomes. Comparison of (**A**) SERBP1 (SERPINE1 mRNA-binding protein 1) and eEF2 (eukaryotic elongation factor 2) bound to the rabbit ribosome with (**B**) Stm1 bound to the yeast ribosome (PDB 4V88). (**C**) Path of SERBP1 and (**D**) Stm1 through the mRNA entrance channel. (**E**) Interaction between SERBP1 and eEF2.

DOI: https://doi.org/10.7554/eLife.40486.004

The following figure supplements are available for figure 2:

**Figure supplement 1.** Distinct classes of eEF2 and SERBP1 bound to 80S ribosomes.

DOI: https://doi.org/10.7554/eLife.40486.005

**Figure supplement 2.** Sequence conservation of Stm1 and SERBP1.

DOI: https://doi.org/10.7554/eLife.40486.006

IFRD2-containing ribosomes from HEK293T cells, which confirmed that IFRD2 binds human ribosomes identically as those observed in the reticulocyte sample (*Figure 3—figure supplement 3F*).

## A new tRNA-binding site on the mammalian ribosome

All ribosomes bound to IFRD2 in the reticulocyte dataset also contain a tRNA in an extreme position on the ribosome past the E site that we call the 'Z site' (*Figure 3A*). Although we could not distinguish any IFRD2-containing reticulocyte ribosomes lacking Z-site tRNA, we do not observe Z-site tRNA in the reconstruction of IFRD2 on the human ribosome (*Figure 3—figure supplement 3F*), indicating that Z-site tRNA is not required for IFRD2 binding. In the combined dataset of reticulocyte ribosomes, Z-site tRNA also binds 80S ribosomes alone and with peptidyl-tRNA (*Figure 1*) but would clash with an E-site tRNA (*Figure 4A*).

Like the universal tRNA-binding positions, Z-site tRNA bridges the ribosomal subunits. Three distinct interactions are observed between the tRNA and the large subunit. First, the backbone of the 3′ CCA binds a lysine-rich stretch of eL42 (residues 27 – 30) (*Figure 4B*). Second, the acceptor arm binds the backbone of helix 68 of 28S rRNA (nucleotides 3730 – 3732), and third, the elbow of the Z-site tRNA interacts with the L1 stalk (*Figure 4B*). At the head of the 40S subunit, the anticodon stem-loop of the Z-site tRNA binds a positively charged site on the non-essential ribosomal protein, eS25 (*Figure 4C*). Although eS25 has not been shown to bind tRNAs previously, it is a common binding site for internal ribosome entry site (IRES) sequences utilized by many viruses, including human immunodeficiency virus (HIV) and the hepatitis C virus (*Quade et al., 2015*; *Yamamoto et al., 2015*), to initiate translation of viral RNA on host ribosomes. The Z-site tRNA most closely resembles the ribosome-binding conformation of the cricket paralysis virus (CPrV) IRES (*Fernández et al., 2014*), which also bridges eS25 and the ribosomal L1 stalk (*Figure 4D*). Thus, viruses appear to exploit the Z site for IRES-mediated translation.

## Discussion

Our structural analysis of reticulocyte ribosomes in a large cryo-EM dataset has revealed new insights into ribosome inactivation in mammals. We identify new similarities between yeast Stm1 and mammalian SERBP1, reveal the interactions that sequester eEF2 on the ribosome, and redefine the IFRD family as ribosome-silencing proteins.

As proposed for Stm1, IFRD2 may repress translation during cellular stress. Previous studies indicate that IFRD2 expression is induced in specific circumstances, such as tetradecanoyl phorbol

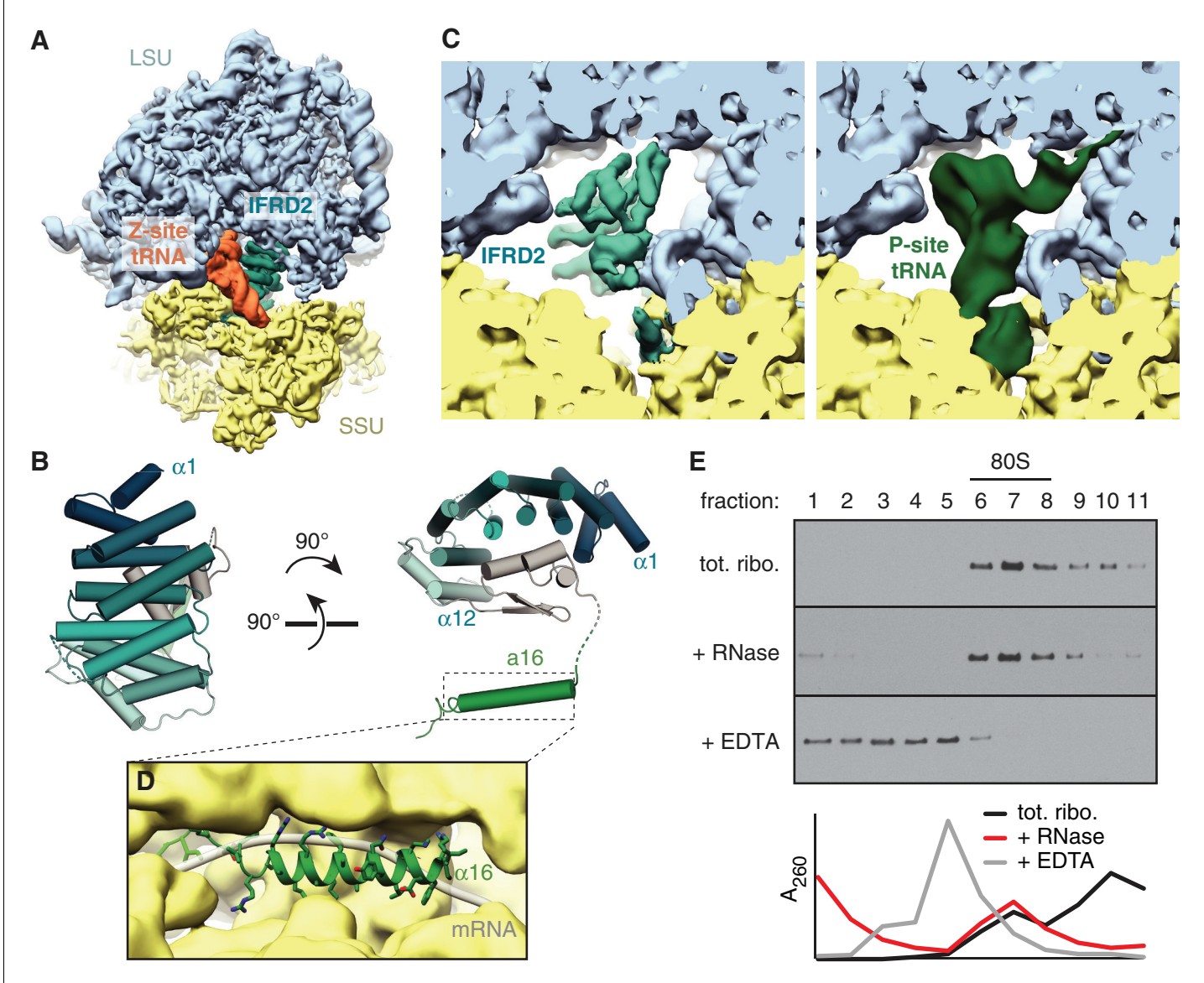

**Figure 3.** IFRD2 binding to ribosomes is incompatible with translation. (**A**) IFRD2 (interferon-related developmental regulator 2) occupies the intersubunit space in a subset of ribosomes containing Z-site tRNA. (**B**) Model of IFRD2 colored by domain. (**C**) IFRD2 interacts with many of the same ribosomal elements as P-site tRNA. (**D**) The C-terminal α-helix of IFRD2 occupies the mRNA exit channel and follows the path taken by mRNA through the ribosome. (**E**) 200 nM ribosomes isolated from rabbit reticulocyte lysate were treated without or with 50 μg/mL RNase A or 10 mM EDTA for 5 min and subjected to native size fractionation on 10 – 50% sucrose gradients. Eleven fractions taken from the top were collected and subjected to immunoblotting for IFRD2. Absorbance readings at 254 nm are shown below.

DOI: https://doi.org/10.7554/eLife.40486.007

The following figure supplements are available for figure 3:

**Figure supplement 1.** IFRD2 interactions with 80S ribosomes.
DOI: https://doi.org/10.7554/eLife.40486.008

**Figure supplement 2.** Sequence alignment of IFRD family members of selected eukaryotes.
DOI: https://doi.org/10.7554/eLife.40486.009

**Figure supplement 3.** Characteristics of IFRD2 association with mammalian ribosomes.
DOI: https://doi.org/10.7554/eLife.40486.010

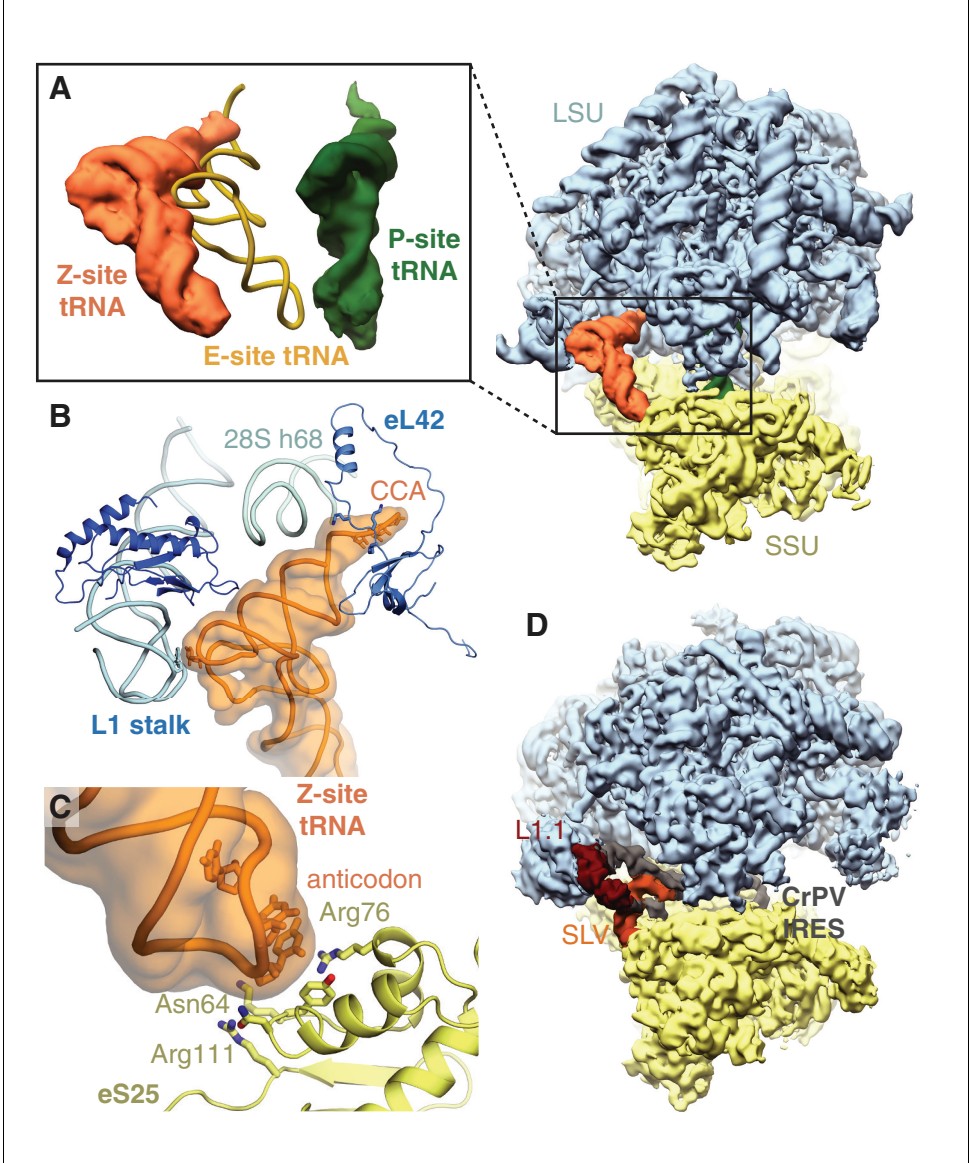

**Figure 4.** Z-site tRNA. (**A**) Position of Z-site tRNA on the ribosome in the presence of a P-site tRNA. Docked E-site tRNA (shown in cartoon representation) is incompatible with a Z-site tRNA. (**B**) Interactions between the Z-site tRNA (CCA nucleotides shown in stick representation) and the 60S ribosomal subunit. (**C**) Interaction between the anticodon stem-loop (anticodon nucleotides shown in stick representation) of Z-site tRNA and eS25. (**D**) Cryo-EM map of CrPV IRES (EMD-2599), with the L1.1 and stem-loop V (SLV) domains, which make similar interactions as Z-site tRNA, indicated, bound to the *Kluyveromyces lactis* ribosome. LSU – large ribosomal subunit; SSU – small ribosomal subunit; IRES – internal ribosome entry site; SLV – stem-loop V.

DOI: https://doi.org/10.7554/eLife.40486.012

acetate treatment and serum replenishment after starvation (*Varnum et al., 1989*). In addition, IFRD proteins appear to escape translational repression by stress-induced eIF2α phosphorylation via an upstream open-reading frame (uORF) (*Andreev et al., 2015*; *Zhao et al., 2010*). IFRD proteins may also provide a general mechanism to reduce translational output, which may explain the pleiotropic effects of IFRD ablation on the differentiation of various cell types (*Vietor et al., 2002*).

Our results suggest that IFRD2 is especially abundant on ribosomes in reticulocytes, the immediate precursors to red blood cells. Reticulocytes also express enhanced levels of other factors that can suppress protein synthesis, including the heme-dependent eIF2α kinase HRI (*Crosby et al., 1994*). Considering the importance of ribosomal levels (*Khajuria et al., 2018*) and global

translational regulation (*Grevet et al., 2018*) on red blood cell differentiation and function, IFRD2 may offer an additional layer of translational control to dictate blood cell fate.

While SERBP1 and IFRD2 may act redundantly to inactivate mammalian ribosomes, our structures suggest they target different subsets of ribosomes and may induce different fates. In yeast, Stm1-inactivated ribosomes are recycled after starvation by Dom34-Hbs1 (Pelota-Hbs1l in mammals) and Rli1 (ABCE1 in mammals) (*van den Elzen et al., 2014*). Whether IFRD2-silenced ribosomes are similarly rescued, and the physiological conditions and timescales of these processes, remain to be determined. The regulation of these two mechanisms may impact their role on global translation activity during different types of cellular differentiation and stress.

We additionally observe Z-site tRNA on all reticulocyte ribosomes containing IFRD2. Although we cannot rule out the possibility that a tRNA in the Z site results from an artefact of isolating these samples, the stable association of Z-site tRNA and the similarity with IRES interactions supports the idea that the Z site is a preferred binding site for certain RNA structures, potentially including cellular mRNAs. In yeast, knocking out eS25, a key interacting partner of Z-site tRNA, decreases global protein synthesis and slows growth (*Ferreira-Cerca et al., 2005*; *Landry et al., 2009*). A recent study also suggests that eS25 is expressed at substoichiometric levels relative to total ribosomes in mammals, and that ribosomes containing eS25 may preferentially translate subsets of mRNAs (*Shi et al., 2017*).

The Z-site may represent a late-stage intermediate of tRNA ejection downstream of the E site. This hypothesis is supported by the observation that Z-site tRNA can bind simultaneously with a P-site tRNA but not a canonical E-site tRNA, and that Z-site tRNA interacts with the L1 stalk, which interacts with all known tRNA-ejection intermediates in bacteria (*Agrawal et al., 2000*) (*Zhang et al., 2018*). However, the interaction of Z-site tRNA with the eukaryotic-specific proteins eL42 and eS25 suggests that the Z site is distinct from bacterial tRNA-ejection intermediates.

A Z-site tRNA ejection intermediate would be transient during active translation, occurring only before the next translocation event. Therefore, the presence of a stably bound Z-site tRNA could act as signature of a translationally incompetent ribosome regardless of whether it engages the Z site from the E site or from a pool of deacylated tRNAs in the cytosol. In this scenario, tRNA binding to the Z site is more likely when deacylated tRNA levels rise, for example by amino acid depletion, or when translational factors are limiting. Intriguingly, deacylated tRNAs, a yeast-specific tRNA ejection factor, and stalled ribosomes are all implicated in regulating the integrated stress response by the GCN2 kinase (*Ishimura et al., 2016*; *Ramirez et al., 1991*; *Visweswaraiah et al., 2012*).

Our unexpected observation of IFRD2 and a new tRNA position on the mammalian ribosome highlights the ability of mining cryo-EM datasets to reveal new biological interactions from heterogeneous samples. This strategy is directly applicable to identifying and visualizing binding partners of other biological assemblies.

## Materials and methods

### Key resources table

| Reagent type (species) or resource | Designation | Source or reference | Identifiers | Additional information |
|---|---|---|---|---|
| Cell Line (*H. sapiens*) | HEK293T | American Type Culture Collection | Cat# CRL-3216 RRID:CVCL_0063 | |
| Recombinant DNA reagent | pcDNA3.1-3x FLAG-TEV-IFRD2 | This paper | N/A | Mammalian expression vector expressing Flag-tagged IFRD2 behind a CMV promoter |
| Antibody | IFRD2 | Invitrogen | Cat# PA5-48833 RRID:AB_2634289 | IB: 1:1000 |
| Antibody | FLAG M2 | Sigma-Aldrich | Cat# F3165 RRID:AB_259529 | IB: 1:5000 |
| Antibody | uL2 | Abcam | Cat# ab169538 RRID:AB_2714187 | IB: 1:10000 |
| Antibody | eS24 | Abcam | Cat# ab196652 RRID:AB)2714188 | IB: 1:3000 |

*Continued on next page*

*Continued*

| Reagent type (species) or resource | Designation | Source or reference | Identifiers | Additional information |
|---|---|---|---|---|
| Antibody | HRP anti-mouse | Jackson ImmunoResearch | Cat# 115-035-003 RRID:AB_10015289 | IB: 1:5000 |
| Antibody | HRP anti-rabbit | Jackson ImmunoResearch | Cat# 111-035-003 RRID:AB_2313567 | IB: 1:5000 |
| Antibody | FLAG M2 agarose resin | Sigma-Aldrich | Cat# A2220 RRID:AB_10063035 | |
| Peptide, recombinant protein | 3X FLAG peptide | Sigma-Aldrich | Cat# F4799 | |
| Peptide, recombinant protein | RNase A | Sigma-Aldrich | Cat# R6513 | |
| Chemical compound, drug | Dulbecco's Modified Eagle Medium (DMEM) | Gibco/Thermo Fisher | Cat# 10569 | |
| Chemical compound, drug | HyClone Fetal Bovine Serum | GE Healthcare Life Sciences | Cat# SH30910.03 | |
| Chemical compound, drug | Trypsin-EDTA | Gibco/Thermo Fisher | Cat# 25200 | |
| Chemical compound, drug | OptiMEM | Gibco/Thermo Fisher | Cat# 31985 | |
| Chemical compound, drug | TransIT 293 | Mirus | Cat# MIR2705 | |
| Chemical compound, drug | EDTA | GrowCells | Cat# MRGF-1202 | |
| Chemical compound, drug | SuperSignal West Pico | Thermo Fisher | Cat# 34080 | |
| Biological sample (*O. cuniculus*) | Rabbit reticulocyte lysate | Green Hectares | N/A | |
| Software, algorithm | RELION-2.0 or RELION-2.1 | *Kimanius et al., 2016* | RRID:SCR_016274 | https://www2.mrc-lmb.cam.ac.uk/relion |
| Software, algorithm | MotionCor2 | *Zheng et al., 2017* | RRID:SCR_016499 | http://msg.ucsf.edu/em/software/motioncor2.html |
| Software, algorithm | GCTF | *Zhang, 2016* | RRID:SCR_016500 | https://www.mrc-lmb.cam.ac.uk/kzhang/ |
| Software, algorithm | UCSF Chimera | *Pettersen et al., 2004* | RRID:SCR_004097 | https://www.cgl.ucsf.edu/chimera/ |
| Software, algorithm | Coot v.0.8.9 | *Brown et al., 2015a* | RRID:SCR_014222 | https://www2.mrc-lmb.cam.ac.uk/personal/pemsley/coot/ |
| Software, algorithm | I-TASSER | *Zhang, 2008* | RRID:SCR_014627 | https://zhanglab.ccmb.med.umich.edu/I-TASSER/ |
| Software, algorithm | PHENIX | *Adams et al., 2010* | RRID:SCR_014224 | https://www.phenix-online.org/ |
| Software, algorithm | MolProbity v.4.3.1 | *Chen et al., 2010* | RRID:SCR_014226 | http://molprobity.biochem.duke.edu/ |
| Software, algorithm | EMRinger | *Barad et al., 2015* | N/A | http://emringer.com/ |
| Software, algorithm | SBGrid | *Morin et al., 2013* | RRID:SCR_003511 | https://sbgrid.org/ |
| Software, algorithm | PyMOL | *DeLano, 2002* | RRID:SCR_000305 | http://www.pymol.org |
| Software, algorithm | SCIPION | *de la Rosa-Trevín et al., 2016* | N/A | http://scipion.i2pc.es/ |
| Software, algorithm | MonoRes | *Vilas et al., 2018* | N/A | |

## Plasmids and antibodies

The ORF of IFRD2 was subcloned into a pcDNA3.1-based vector after a CMV promoter and an N-terminal 3X Flag tag and TEV cleavage site. IFRD2 antibody was obtained from Invitrogen (PA5-48833). Anti-FLAG M2 monoclonal antibody (F3165), resin (A2220) and 3X FLAG peptide (F4799) were obtained from Sigma-Aldrich.

## Mass spectrometry

Three samples for mass-spectrometry analysis were prepared to identify the proteins present in the samples used for cryo-EM (*Supplementary file 1*). The first was prepared exactly as for cryo-EM analysis (*Brown et al., 2015b*). Briefly, we performed 2 mL in vitro translation reactions of a model substrate encoding an N-terminal 3X Flag tag, the autonomously folding villin headpiece domain, and the cytosolic portion of Sec61β in the presence of 0.5 µM dominant negative eRF1 in which the catalytic GGQ motif is mutated to AAQ to trap ribosomes at stop codons. Immediately after the translation reaction, we affinity purified the stalled ribosome-nascent protein complexes via the N-terminal 3X Flag tag on the nascent protein using anti-Flag M2 agarose beads (Sigma). The reactions were incubated with M2 affinity resin at 4°C for 1 hr, followed by three sequential 6 mL washes with: (1) 50 mM Hepes pH 7.4, 100 mM KOAc, 5 mM Mg(OAc)$_2$, 0.1% Triton X-100, 1 mM DTT, (2) 50 mM Hepes pH 7.4, 250 mM KOAc, 5 mM Mg(OAc)$_2$, 0.5% Triton X-100, 1 mM DTT, and (3) 50 mM Hepes pH 7.4, 100 mM KOAc, 5 mM Mg(OAc)$_2$, 1 mM DTT. Two sequential elutions were performed with 0.1 mg/mL 3X Flag peptide in the third wash buffer for 25 min each at room temperature. This elution was directly analyzed as 'sample A' (*Supplementary file 1*).

To reduce the identification of ribosomal protein peptides, we evenly divided the elutions into two halves. We maintained one half in physiological salt concentrations and adjusted the other half to a final concentration of 750 mM KOAc, 15 mM Mg(OAc)$_2$. We centrifuged both samples at 100,000 rpm for 30 min in a TLA120.2 rotor to pellet ribosomes and strip off peripherally associated proteins under the high-salt conditions. The supernatants from both spins were precipitated with trichloroacetic acid (TCA) and analyzed by SDS-PAGE and Coomassie staining. The entire gel lanes were excised and submitted for mass spectrometry analysis as 'sample B' (physiological salt wash) and 'sample C' (high-salt wash). The proteins identified by mass-spectrometry are given in *Supplementary file 1*.

## Cell, lysate, and ribosomal treatments

HEK293T cells were maintained in Dulbecco's Modified Eagle Medium (DMEM) with high glucose and 10% fetal bovine serum and verified to be mycoplasma free. Transfections of Flag-tagged IFRD2 were performed when cells were 60 – 70% confluency using TransIT 293 (MIRUS) according to manufacturer instructions. 16 – 18 hr after transfection, we lysed cells in 50 mM Hepes pH 7.4, 100 mM KOAc, 5 mM Mg(OAc)$_2$, 0.5% Triton X-100, 1 mM DTT, and protease inhibitor cocktail. Lysates were clarified by spinning at 20,000xg at 4°C for 10 min before being subjected to biochemical treatments.

Where applicable, rabbit reticulocyte lysate (Green Hectares; *Figure 3—figure supplement 3A*), 200 nM ribosomes (*Figure 3E*) isolated from reticulocytes or HEK293T cell lysate (*Figure 3—figure supplement 3C*) were treated with 50 µg/mL RNase A at 25°C for 5 min before sucrose gradients. RNase A efficiency was confirmed after sucrose gradients by measuring absorbance at 254 nm to confirm the collapse of polysomes into monosome fractions (*Figure 3E*). EDTA was included with ribosomes isolated from reticulocytes (*Figure 3E*) or ribosomes pelleted after Flag-tagged IFRD2 affinity purification (*Figure 3—figure supplement 3E*) at a final concentration 10 mM. All size fractionation were performed using 200 µL 10–50% sucrose gradients prepared in 50 mM Hepes pH 7.4, 100 mM KOAc, 5 mM Mg(OAc)$_2$ without or with 10 mM EDTA.

Samples were layered on top of the gradients and spun at 55,000 rpm using a TLS-55 rotor in an OptimaMax ultracentrifuge (Beckman Coulter) for 30 min at 4°C using the slowest acceleration and deceleration settings. Eleven 20 µL fractions were manually collected from the top for immunoblotting analysis.

## Affinity purification and cryo-EM analysis of human IFRD2-80S ribosome complexes

Four 10 cm plates of HEK293T cells were transfected as described above and passaged 1:4 the day after transfection. Two days later, the cells were lysed as described above and incubated with M2 anti-Flag agarose resin for 1 hr at 4°C. The resin was washed with 6 mL of lysis buffer and subjected to two sequential elutions using one column volume of 0.1 mg/mL 3X Flag peptide incubated at room temperature for 25 min. A portion of the elution was directly frozen onto cryo-EM grids, and the leftover elution was pelleted at 100,000 rpm for 40 min at 4°C in a TLA120.2 rotor to isolate ribosomes, which were subjected to biochemical analysis (*Figure 3—figure supplement 3E*).

For cryo-EM analysis, 3 µL of the pooled elution containing ~80 nM ribosomes were directly frozen onto glow-discharged Quantifoil Cu R2/2 grids with a thin (~50 Å) continuous layer of carbon using a Vitrobot Mark III at 4°C and 100% humidity with 30 s wait time and 3 s blot time. A dataset of 877 micrographs was collected using an automated data collection pipeline in SerialEM on a Talos Arctica operated at 200 kV. Images were acquired at a nominal magnification of 36,000x (corresponding to a pixel size of 1.169 Å) with a Gatan K2 direct electron detector in super-resolution mode and defocus values ranging from $-1.3$ to $-3$ µm. Each movie was acquired at 5 frames/s over a total exposure time of 8 s and a dose rate of 5.4 electrons/pixel/s.

## Data processing

To take advantage of recent developments in cryo-EM image processing, we reprocessed datasets of ribosomes isolated in the presence of a dominant-negative eukaryotic release factor (DN-eRF1) (*Brown et al., 2015b*). All processing steps were performed within RELION-2.0 or RELION-2.1 (*Kimanius et al., 2016*). We used MotionCor2 (*Zheng et al., 2017*) to correct for global and local (5 × 5 patches) beam-induced motion and to dose weight the individual frames. The motion-corrected sums without dose weighting were used for CTF estimation with GCTF (*Zhang, 2016*). All motion- and CTF-corrected micrographs and their Fourier transforms were inspected manually to remove those that displayed astigmatism or ice contamination.

To generate reference templates for auto-picking, 2000 ribosomal particles were picked manually and subjected to 2D classification. The best five classes (out of 10) were low-pass filtered to 20 Å and used for auto-picking in RELION (*Scheres, 2015*). For autopicking, a mask diameter of 360 Å and a minimum interparticle distance of 200 Å was used. The autopicked particles were extracted with a box size of 400 pixels and sorted by a Z-score generated by subtracting the reference image from the extracted particles. The sorted particles were inspected manually and non-ribosomal particles with a low Z score discarded. Retained particles were subjected to reference-free 2D classification and the best-resolved classes selected. In total, 971,191 particles were retained after 2D classification.

All selected particles were subjected to an initial round of refinement using RELION's 3D autorefine in which a 30 Å low-pass filtered map of the rabbit 80S ribosome with tRNAs but no factors (EMD-4129) (*Shao et al., 2016*) was used as a reference. Following refinement, a single round of three-dimensional classification without alignment was performed. This strategy was very effective at separating rotated and unrotated 80S ribosomes from 60S ribosomal subunits.

To isolate particles corresponding to different classes within the set of 80S particles we used multiple rounds of focused classification with signal subtraction (FCwSS) (*Bai et al., 2015*) centered on the A, P, E, and Z sites of the ribosome. These sites were chosen as they displayed nebulous density after the initial round of refinement, consistent with mixed occupancy. The use of FCwSS in classifying the data represents the most significant difference from how the maps were processed originally (*Brown et al., 2015b*).

Once a defined class had been isolated, the particles were realigned, and the resultant maps postprocessed in RELION. For post-processing, solvent masks were generated using relion_mask_create. Typically, these binary masks were generated from the final map from refinement low-pass filtered to 15 Å at a threshold of 0.015 or 0.02. The initial binary mask was extended by 4 Å in all directions and a raised-cosine edge was added to create a soft mask. During post-processing, phase-randomization was used to correct for the convolution effects of the solvent mask. Overall resolution estimates were calculated from Fourier shell correlations at 0.143 between the two

independently refined half-maps. Final reconstructions were sharpened using automatically estimated B-factors (*Rosenthal and Henderson, 2003*).

Processing of the human IFRD2-80S ribosomal complex was performed in RELION 2.1 as described above. After 2D classification, 22,807 particles were subject to an initial refinement using EMD-4129 low-pass filtered to 40 Å as a reference. We used the refined map, which already had clear IFRD2 density, as a reference for 3D classification to isolate 80S ribosomes, followed by a round of FCwSS centered on IFRD2 density. A final refinement of 5714 particles resulted in an 8.3 Å map without postprocessing.

## Model building and refinement

Our 3.6 Å map of 80S•eRF1•ABCE1 complex was used as the starting point for model building. The model of the rabbit 80S ribosome in an unrotated state (PDB accession code 5LZV) (*Shao et al., 2016*) was fitted into this map using the 'fit in map' feature of Chimera (*Pettersen et al., 2004*). All non-ribosomal elements were deleted, and chemical modifications added to ribosomal RNA and proteins uL4 and eL40 using the cryo-EM structure of the human ribosome as a guide (*Natchiar et al., 2017*). This model was propagated to all other maps as individual 60S and 40S subunits. For the rotated and head-swiveled classes, the individual proteins and rRNA in the head of the 40S subunit were fitted as rigid bodies following docking of the ribosomal subunits.

Protein factors and tRNAs were modeled in Coot v0.8.9 (*Brown et al., 2015a*). P- and E-site tRNAs were extracted from the rabbit 80S ribosome (PDB accession code 5LZV) (*Shao et al., 2016*). The Z-site tRNA was modeled using the same sequence as for the E-site tRNA, although the density comes from a mixture of different tRNAs. The model for the L1 stalk was built using the crystal structure of the L1-stalk fragment from *Haloarcula marismortui* (PDB accession code 5Ml7) as a template. The rabbit L1 protein (NCBI ID: XP_002714675) was modeled by docking a comparative model generated using I-TASSER (*Zhang, 2008*) and morphing it to fit the density using Phenix.real_space_refinement (*Afonine et al., 2018*). The model for IFRD2 (NCBI XP_002713258) was built de novo from polyalanine helices placed into the map. Sequences were assigned to these helices based on side chain density and the helices connected by manual model building in Coot. eEF2 was modeled using the model of *Sus scrofa* eEF2 (PDB accession code 3J7P) (*Voorhees et al., 2014*) as a template. Human numbering and sequence (Uniprot ID: P13639) was used as sequence information is not available for rabbit eEF2. SERBP1 isoform X2 (XP_002715981) was built de novo into the density.

All models were refined using Phenix.real_space_refinement v1.13_2998 (*Afonine et al., 2018*). Each round of global real-space refinement featured five macro-cycles with secondary structure, rotamer, Ramachandran, and Cβ-torsion restraints applied. Secondary structure restraints were determined directly from the model and recalculated for each round of refinement. For the rRNA and tRNAs present in the models, hydrogen-bonding and base-pair and stacking parallelity restraints were applied. Additional restraints were applied for the chemical modifications of the ribosome and the diphthamide modification of eEF2. These restraints were generated using phenix.readyset. The high-resolution limit was set during refinement to match the nominal resolution obtained by postprocessing in Relion.

The final models were validated using MolProbity v.4.3.1 (*Chen et al., 2010*) and EMRinger (*Barad et al., 2015*), with final statistics given in table S2. Over-fitting was monitored using cross-validation (*Amunts et al., 2014*) (*Figure 1—figure supplement 1*).

Software used in the project were installed and configured by SBGrid (*Morin et al., 2013*).

## Figures

Figure panels were generated using PyMOL (*DeLano, 2002*) or Chimera (*Pettersen et al., 2004*). Maps colored by local resolution were generated with unsharpened density maps using MonoRes (*Vilas et al., 2018*).

## Acknowledgements

The authors thank V Ramakrishnan and RS Hegde, M Skehel and the MRC-LMB mass-spectrometry facility, Harvard Research Computing and SBGrid for computing support, E Fischer and SHarrison for access to their GPU workstations, the cryo-EM facility at the University of Massachusetts Worcester for help with cryo-EM data collection, L Hollingsworth and T Rapoport for comments, the Shao

lab for helpful discussions, and NVIDIA Corporation for the donation of a Quadro P6000 GPU. This work was supported by Harvard Medical School, the International Retinal Research Foundation, the E Matilda Ziegler Foundation, the Richard and Susan Smith Family Foundation, and the Charles H Hood Foundation.

## Additional information

### Funding

| Funder | Author |
|---|---|
| Harvard Medical School | Alan Brown<br>Matthew R Baird<br>Matthew CJ Yip<br>Sichen Shao |
| International Retinal Research Foundation | Alan Brown |
| E. Matilda Ziegler Foundation for the Blind | Alan Brown |
| Eunice Kennedy Shriver National Institute of Child Health and Human Development | Jason Murray |
| Charles H. Hood Foundation | Sichen Shao |
| Richard and Susan Smith Family Foundation | Sichen Shao |

The funders had no role in study design, data collection and interpretation, or the decision to submit the work for publication.

### Author contributions

Alan Brown, Sichen Shao, Conceptualization, Supervision, Investigation, Writing—original draft, Writing—review and editing; Matthew R Baird, Matthew CJ Yip, Jason Murray, Investigation, Writing—review and editing

### Author ORCIDs

Alan Brown http://orcid.org/0000-0002-0021-0476
Matthew CJ Yip http://orcid.org/0000-0002-2505-9987
Jason Murray http://orcid.org/0000-0002-8679-8281
Sichen Shao http://orcid.org/0000-0003-2679-5537

### Decision letter and Author response

Decision letter https://doi.org/10.7554/eLife.40486.042
Author response https://doi.org/10.7554/eLife.40486.043

## Additional files

### Supplementary files

• Transparent reporting form
DOI: https://doi.org/10.7554/eLife.40486.014

• Supplementary file 1. Curated list of the proteins observed by mass spectrometry used to identify factors in the cryo-EM maps. For clarity, ribosomal proteins and contaminating bacterial and skin proteins are excluded and known components of multisubunit complexes clustered at the end of the table. Sample A represents the sample used for cryo-EM. Sample B are the proteins eluted from ribosomes under physiological salt conditions. Sample C are the proteins eluted from ribosomes under high-salt (750 mM KOAc, 15 mM Mg(OAc)$_2$) conditions. Proteins observed in cryo-EM complexes are highlighted in yellow.
DOI: https://doi.org/10.7554/eLife.40486.015

## Data availability

All cryo-EM maps and models have been deposited in EMDB under accession codes 9234, 9235, 9236, 9237, 9239, 9240, 9241 and 9242. All models have been deposited in PDB under accession codes 6MTB, 6MTC, 6MTD and 6MTE.

The following datasets were generated:

| Author(s) | Year | Dataset title | Dataset URL | Database and Identifier |
|---|---|---|---|---|
| Brown A, Baird MR, Yip MCJ, Murray J, Shao S | 2018 | Rabbit 80S ribosome with a P-site tRNA (unrotated state) | http://www.ebi.ac.uk/pdbe/entry/emdb/EMD-9234 | EMBL-EBI Protein Data Bank, EMD-9234 |
| Brown A, Baird MR, Yip MCJ, Murray J, Shao S | 2018 | Rabbit 80S ribosome with a P- and E-site tRNA | http://www.ebi.ac.uk/pdbe/entry/emdb/EMD-9235 | EMBL-EBI Protein Data Bank, EMD-9235 |
| Brown A, Baird MR, Yip MCJ, Murray J, Shao S | 2018 | Rabbit 80S ribosome with a Z-site tRNA (unrotated state) | http://www.ebi.ac.uk/pdbe/entry/emdb/EMD-9236 | EMBL-EBI Protein Data Bank, EMD-9236 |
| Brown A, Baird MR, Yip MCJ, Murray J, Shao S | 2018 | Rabbit 80S ribosome with P- and Z-site tRNAs (unrotated state) | http://www.ebi.ac.uk/pdbe/entry/emdb/EMD-9237 | EMBL-EBI Protein Data Bank, EMD-9237 |
| Brown A, Baird MR, Yip MCJ, Murray J, Shao S | 2018 | Rabbit 80S ribosome with Z-site tRNA and IFRD2 (unrotated state) | http://www.ebi.ac.uk/pdbe/entry/emdb/EMD-9239 | EMBL-EBI Protein Data Bank, EMD-9239 |
| Brown A, Baird MR, Yip MCJ, Murray J, Shao S | 2018 | Rabbit 80S ribosome with eEF2 and SERBP1 (unrotated state with 40S head swivel) | http://www.ebi.ac.uk/pdbe/entry/emdb/EMD-9240 | EMBL-EBI Protein Data Bank, EMD-9240 |
| Brown A, Baird MR, Yip MCJ, Murray J, Shao S | 2018 | Rabbit 80S ribosome with A/P and P/E tRNAs (rotated state) | http://www.ebi.ac.uk/pdbe/entry/emdb/EMD-9241 | EMBL-EBI Protein Data Bank, EMD-9241 |
| Brown A, Baird MR, Yip MCJ, Murray J, Shao S | 2018 | Rabbit 80S ribosome with eEF2 and SERBP1 (rotated state) | http://www.ebi.ac.uk/pdbe/entry/emdb/EMD-9242 | EMBL-EBI Protein Data Bank, EMD-9242 |
| Brown A, Baird MR, Yip MCJ, Murray J, Shao S | 2018 | Rabbit 80S ribosome with P- and Z-site tRNAs (unrotated state) | http://www.rcsb.org/pdb/explore/explore.do?structureId=6MTB | RCSB Protein Data Bank, 6MTB |
| Brown A, Baird MR, Yip MCJ, Murray J, Shao S | 2018 | Rabbit 80S ribosome with Z-site tRNA and IFRD2 (unrotated state) | http://www.rcsb.org/pdb/explore/explore.do?structureId=6MTC | RCSB Protein Data Bank, 6MTC |
| Brown A, Baird MR, Yip MCJ, Murray J, Shao S | 2018 | Rabbit 80S ribosome with eEF2 and SERBP1 (unrotated state with 40S head swivel) | http://www.rcsb.org/pdb/explore/explore.do?structureId=6MTD | RCSB Protein Data Bank, 6MTD |
| Brown A, Baird MR, Yip MCJ, Murray J, Shao S | 2018 | Rabbit 80S ribosome with eEF2 and SERBP1 (rotated state) | http://www.rcsb.org/pdb/explore/explore.do?structureId=6MTE | RCSB Protein Data Bank, 6MTE |

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
