## [Decision Letter]

Thank you for submitting your article "Structures of translationally inactivated mammalian ribosomes" for consideration by *eLife*. Your article has been reviewed by James Manley as the Senior Editor, a Reviewing Editor and three peer reviewers. The reviewers have opted to remain anonymous.

The reviewers have discussed the reviews with one another and the Reviewing Editor has drafted this decision to help you prepare a revised submission.

Summary:

The authors report cryo-EM structures of mammalian 80S ribosomes bound with SERBP1•eEF2 or with IFRD2. Importantly, they document a tRNA-binding, which differs from the canonical A, P and E sites. The structures allow near-atomic interpretation of interactions between the ribosomes, SERBP1 and eEF2, shedding new light on the mechanism of SERBP1. The work describes several new findings that will be of interest to researchers in and beyond the ribosome field.

Essential revisions:

The reviewers believe that your paper contains novel and interesting data, but that the paper needs revisions to (1) include reference to biochemical/biophysical work in the field (please see in the reviews below) and (2) provide a clearer explanation/review of the biological relevance of these structures as "inactivated" complexes. As you can see reviewer #1, in particular, is concerned whether the "inactive" states are just an artifact of isolation and treatment or whether they represent true inactive states that might reflect different physiological conditions. Also, there is the possibility that they represent intermediates (i.e. perhaps during recycling)?

*Reviewer #1:*

This is an interesting manuscript that reports on the structure of complexes of "inactivated" ribosomes. As such, these may offer little new insight into the process of translation but may be very important in elements relating to the regulation of translation and availability of ribosomal subunits. A major concern for this paper is whether the structures determined are biologically relevant as these inactive complexes have not been shown to directly be related to any process within the cell that might have generated such inactive particles (i.e. either as soluble complexes or as stress granules). Secondly, in the partial purification of ribosomal complexes, the authors use high speed centrifugation which is known to cause inactivation of ribosomes (Lu et al., 1997 or Scheck and Landau, 1982). Thus, there is the concern as to whether the complexes observed represent native structures or ones altered by hydrostatic pressure. However, that said, it is impressive that the authors can resolve so many structures from a large dataset and offers promise for the ability to readily obtain almost any ribosomal structure.

1) Title – it is not clear that these ribosomes have been inactivated, but rather that they are in an inactive state as the A, P and E sites or mRNA channel are blocked. There is the possibility that some of these complexes (i.e. 80S•P•E•eRF1•ABCE1) may represent intermediates in the translation cycle (and therefore, should not be considered inactive).

2) The authors use affinity tags for purification, but it is not clear that this process has actually worked. How is this population different from one where no affinity purification is used?

3) It is anticipated that a comparison of the yield of different proteins as determined by mass spectrometry should be reflected in the types of particles that are examined. Is there any data that would allow one to relate the amount of protein (as reflected in Supplementary file 1 to the different ribosomal classes as identified in Table 1)?

4) Supplementary file 1 – Initiation factors: the authors should identify the subunits of eIF2 as α, β and γ so as to not confuse the reader with eIF2A, a separate initiation factor (MW 65,000). Secondly, is there any explanation as to why eIF1, 1A, 5 or 5B were not found?

5) IFRD2 – The authors have failed to indicate that rabbit reticulocytes are essentially "interferon treated cells" in that they have elevated levels of several normally interferon induced proteins (PKR, oligo(A) synthase and RNAse L). Thus, this complex is likely to not normally be present in most tissues in the absence of exposure to interferon.

6) For the Z site tRNA, is it possible that this unique conformation is influenced by eIF5A which binds in this vicinity (i.e. would this small protein have been seen?)?

7) Discussion section – "tRNA binding to the Z site is more likely during periods of elevated deacylated tRNA levels.". This is not likely to occur as a slight increase in deacylated tRNA leads to the phosphorylation of eIF2 and the shutoff of protein synthesis (thus sparing the use of aminoacyl-tRNAs).

*Reviewer #2:*

Brown and co-authors describe cryo-EM structures of mammalian 80S ribosomes bound with SERBP1•eEF2 (at a higher resolution than the previously published 80S•SERBP1•eEF2 structural work) or with IFRD2. In addition, a tRNA-binding site is described, which differs from the canonical A, P and E sites. These results were obtained by re-evaluation of a previously published cryo-EM data set. The structures allow near-atomic interpretation of interactions between the ribosomes, SERBP1 and eEF2, bringing insights into the mechanism of SERBP1, however the authors' description of the mechanism is not clear (see below). The 80S•IFRD2 structure is an exciting finding, which identifies a new protein's role in translation regulation. The new tRNA position near the E site echoes a recent work on the bacterial ribosome. The authors propose that this position is on the pathway of deacyl-tRNA dissociation from the E site. Overall this work describes several new findings that will be of interest to researchers in and beyond the ribosome field.

The following changes/clarifications are suggested:

1) The mechanistic implications of SERBP1•eEF2 complex remain unclear. Based on the finding of SERBP1 bound together with eEF2, the authors propose that SERBP1 depletes cellular eEF2. First, it would be helpful to expand the discussion and mention that yeast homologue Stm1 binds yeast ribosomes without eEF2. Do the mechanisms of translation regulation by Stm1 and SERBP1 differ in that the latter depletes not only the 80S but also eEF2? Alternatively, the authors may have missed 80S•SERBP1 particles in their data set. Such particles may even be more abundant than 80S•SERBP1•eEF2 but also more dynamic, thus difficult to average and classify. Second, the mechanistic description of how SERBP1 prevents eEF2 dissociation remains vague. Comparison with other available eEF2 structures (with sordarin or other drugs and/or GDPNP/GDP nucleotides) might bring insights into how the conformational dynamics of eEF2 and/or the ribosome are affected by SERBP1.

2) IFRD2 is proposed to regulate translation by binding to non-translating 80S. But could IFRD2 also bind vacant 40S and prevent translation initiation?

3) Describe the rotational state of the 80S bound with P-tRNA and Z-tRNA (likely non-rotated, but it would help to state this in the paper). Does IFRD2 induce a different 80S conformation?

*Reviewer #3:*

This manuscript describes the results of a refined analysis of a data set of nearly 1 million ribosome particles obtained from an in vitro rabbit reticulocyte translation reaction in which a dominant negative release factor had been added to enrich for translation termination complexes (Brown et al., 2015b). Evidently, individual complexes only represent a small portion (5-25%) of the ribosomes isolated by the procedures used and the authors have mined this reaction for complexes other than those that have already been published to define 9 new structures (described in summary in Figure 1). There is little information in the present manuscript that describes how the complexes examined here were specifically obtained from the reticulocyte lysate, but it would appear that they work under conditions in which only 80S particles form and that the isolated 80S peak is filled with a diverse array of distinct ribosome complexes. In the Brown et al., 2015 paper it appears to describe a FLAG-tag pull down of the nascent peptide, where focus was given to the eRF1-trapped sub-population of complexes. One of the 9 complexes defined here appears to be the same structure as that which was published in Brown et al., 2015b (the eRF1-ABCE1 POST complex), indicating a duplication or positive control of sorts. While I might be able to understand how the other P-site tRNA-containing complexes were isolated by a nascent polypeptide-directed pull-down approach, it is unclear how the complexes lacking P-site tRNA were obtained (i.e. the focus of the present investigations). I would strongly suggest further clarification on these points for the sake of reproducibility.

The two translationally inactive complexes discussed in detail in Figure 2, Figure 3 and Figure 4 are certainly interesting in regards to the mechanisms by which cells may "store" excess ribosomes and the quality of the data appear sufficient enough to support the claims that the authors make, which are largely qualitative descriptions of the SERBP1 and IFRD2 binding sites and the global positioning of the deacylated tRNA within the E site. In this regard, I think the paper is perfectly suitable for publication.

What appears to need further clarification and elaboration is the discussion surrounding the "Z" site for tRNA binding. Alternative positions for E-site tRNA have been described since the early 80s in the early fluorescence studies of eukaryotic ribosomes (Robertson and Wintermeyer, 1987; Rodnina and Wintermeyer, 1992). Movements of deacylated tRNA within the E site have also been specifically described in recent single-molecule fluorescence studies (Ferguson et al., 2015 and references therein). Although the authors state that their "Z" site position may be similar or related to the position (E_out_) described by (Zhang et al., 2017), I don't see how they make this connection. I also don't understand the authors reference to the "known role [of the L1 stalk] in escorting deacylated tRNA from the ribosome", citing two bodies of work in bacteria that make no such claim or data to support such a claim. In this regard, is it really wise to give it a name other than E' or something of the sort? I fear that re-naming it the "Z" site will significantly complicate its description in future publications. The new position the authors see is clearly an "E site" position, yes?

Aside from these major points, the paper certainly seems publishable in its present format from my perspective.

---

## [Author Response]

Reviewer #1:This is an interesting manuscript that reports on the structure of complexes of "inactivated" ribosomes. As such, these may offer little new insight into the process of translation but may be very important in elements relating to the regulation of translation and availability of ribosomal subunits. A major concern for this paper is whether the structures determined are biologically relevant as these inactive complexes have not been shown to directly be related to any process within the cell that might have generated such inactive particles (i.e. either as soluble complexes or as stress granules).

Although SERBP1/Stm1 has been implicated in stress responses (Van Dyke et al., 2006)(Lee et al., 2014), inactive ribosomes may also occur outside periods of stress. Both SERBP1 and IFRD2 are expressed in most tissues at moderate levels (https://www.proteinatlas.org/ENSG00000214706-IFRD2/tissue and https://www.proteinatlas.org/ENSG00000142864-SERBP1/tissue) and are detected to be associated with ribosomes in unstressed cells (Simsek et al., 2017). This suggests that a percentage of ribosomes exist in inactivate states even in cells not undergoing stress. As we mention in the paper, one possibility is that ribosomes are inactivated as part of translational reprogramming that occurs during differentiation. As our ribosomes were purified from reticulocytes, the penultimate stage of erythrocyte differentiation, this may explain the presence of inactive ribosomes in our sample. Consistent with this, we observe higher IFRD2 levels in reticulocytes than in HEK293T cells (Figure 3—figure supplement 3B).

Secondly, in the partial purification of ribosomal complexes, the authors use high speed centrifugation which is known to cause inactivation of ribosomes (Lu et al., 1997 or Scheck and Landau, 1982). Thus, there is the concern as to whether the complexes observed represent native structures or ones altered by hydrostatic pressure. However, that said, it is impressive that the authors can resolve so many structures from a large dataset and offers promise for the ability to readily obtain almost any ribosomal structure.

Though we cannot completely rule out the possibility that high-speed centrifugation may inactivate some reticulocyte ribosomes, we believe there is strong evidence that the structures of inactive ribosomes we analyze represent physiological complexes and are not the result of technical artifacts:

First, the ability to detect known intermediates of translation (Figure 1) demonstrates that our purification scheme is capable to retaining ribosomes in near-native states. In addition, using the same centrifugation conditions as in this manuscript, we can isolate ribosomal pellets that translate just as efficiently in fractionated cell-free translation systems as complete reticulocyte lysate (Shao and Hegde, 2011), suggesting that these centrifugation conditions do not significantly inactivate ribosomes. Similar centrifugation conditions are used to isolate ribosomes to generate other prokaryotic and eukaryotic cell-free translation systems (Spedding, 1990; Shimizu et al., 2001), including the translation system used in Lu et al., 1997. We think this is consistent with the studies by Lu et al., 1997 and Scheck and Landau, 1982, which primarily analyze ribosome activity after applying high-pressure treatments in closed pressure vessels, rather than by centrifugation.

Second, the interaction between IFRD2 and ribosomes is also detected in a mass spectrometry study that did not use high-speed centrifugation (Simsek et al., 2017), as well as in immunoprecipitations of Flag-tagged IFRD2 from 293T cell lysate that was not subject to centrifugation (Figure 3—figure supplement 3D). We also detect endogenous IFRD2 quantitatively associated with ribosomes in reticulocyte lysate before (Figure 3—figure supplement 3A) and after (Figure 3E) highspeed centrifugation. Finally, recombinant IFRD2 only weakly associates with ribosomes after centrifugation when incubated with lysate (Author response image 1), unlike IFRD2 expressed in cells, which clearly interacts with ribosomes (Author response image 1, Figure 3—figure supplement 3B-E). Similarly, the association of SERBP1/Stm1 with ribosomes has also been observed by different methods in the literature (Zinoviev et al., 2015). These observations all suggest that these complexes assemble in physiological conditions and are not induced by centrifugation.

**Author response image 1. respfig1:** Assembly of IFRD2 onto ribosomes. Recombinant IFRD2 purified from *E. coli* was incubated with cytosol isolated from HEK293T cells for 30 minutes at 32°C and subject to native size fractionation by centrifugation on a 10-50% sucrose gradient. Eleven fractions were collected from the top of the gradient and analyzed by SDS-PAGE and immunoblotting for the recombinant IFRD2 (top blot; quantified by blue curve). This revealed that only minor co-association of IFRD2 in ribosomal (80S) fractions. In contrast, transiently transfecting IFRD2 into HEK293T cells, followed by cytosol isolation and size fractionation (bottom plot; quantified by orange curve) reveals a distinct peak of IFRD2 co-association in ribosomal fractions (red arrow).

1) Title – it is not clear that these ribosomes have been inactivated, but rather that they are in an inactive state as the A, P and E sites or mRNA channel are blocked. There is the possibility that some of these complexes (i.e. 80S•P•E•eRF1•ABCE1) may represent intermediates in the translation cycle (and therefore, should not be considered inactive).

As shown in Figure 1 we only consider ribosomal complexes containing SERBP1 and IFRD2 to be inactive. To more accurately describe these complexes, we have changed the title to “Structures of translationally inactive mammalian ribosomes”.

2) The authors use affinity tags for purification, but it is not clear that this process has actually worked. How is this population different from one where no affinity purification is used?

We address this comment in detail in response to reviewer #3.

3) It is anticipated that a comparison of the yield of different proteins as determined by mass spectrometry should be reflected in the types of particles that are examined. Is there any data that would allow one to relate the amount of protein (as reflected in Supplementary file 1 to the different ribosomal classes as identified in Table 1)?

Our mass spectrometry analysis was run for protein detection of SDS-PAGE gel lanes and not quantification (e.g. via multiplexing of different samples for comparison). As proteins respond differently to trypsin digestion, some peptides ionize better than others, and detection efficiencies for ions with different *m/z* values are unequal, the peptides detected may not fully correlate with the protein composition of the sample. We do detect many more proteins by mass spectrometry (Supplementary file 1) than we identified by cryo-EM analysis (Table 1). This may be due to numerous factors, including relative protein abundance, mRNA-binding proteins that do not directly contact the ribosome, or the inability to visualize flexible ribosome-binding proteins.

4) Supplementary file 1 – Initiation factors: the authors should identify the subunits of eIF2 as α, β and γ so as to not confuse the reader with eIF2A, a separate initiation factor (MW 65,000). Secondly, is there any explanation as to why eIF1, 1A, 5 or 5B were not found?

We have edited Supplementary file 1 to include the names of the eIF2 subunits alongside their Uniprot IDs. We note that eIF5B did appear in our mass spectrometry analysis (G1TRL5_RABIT, IF2P_HUMAN). The absence of other initiation factors may be due to their absence from the sample, presence at concentrations below the detection limit, or technical limitations of the mass spectrometry experiment, particularly given the small sizes of eIF1 and 1A.

5) IFRD2 – The authors have failed to indicate that rabbit reticulocytes are essentially "interferon treated cells" in that they have elevated levels of several normally interferon induced proteins (PKR, oligo(A) synthase and RNAse L). Thus, this complex is likely to not normally be present in most tissues in the absence of exposure to interferon.

As we mention in the manuscript, the name “interferon-related developmental regulator” appears to be a misnomer that originated from a mistaken sequence alignment with mouse interferon-b (Tirone et al., 1989). We do not know of a direct connection between IFRD2 expression or function and interferon signaling, though we cannot exclude the possibility that IFRD2, like established interferon-induced proteins, may be upregulated in certain situations to inhibit translation. In addition, IFRD2 is expressed in many tissues (https://www.proteinatlas.org/ENSG00000214706-IFRD2/tissue), and we can also detect low levels of endogenous IFRD2 in HEK293T cells (Figure 3—figure supplement 3B). We therefore think that IFRD2 is a general mechanism of translationally repressing ribosomes.

6) For the Z site tRNA, is it possible that this unique conformation is influenced by eIF5A which binds in this vicinity (i.e. would this small protein have been seen?)?

eIF5A was our first thought when we saw density in this area prior to classification. However, following extensive classification, we see no evidence of eIF5A bound to any ribosomes. eIF5A has been observed in another cryo-EM study (Schmidt et al., 2016)

7) Discussion section – "tRNA binding to the Z site is more likely during periods of elevated deacylated tRNA levels.". This is not likely to occur as a slight increase in deacylated tRNA leads to the phosphorylation of eIF2 and the shutoff of protein synthesis (thus sparing the use of aminoacyl-tRNAs).

Work in yeast has shown that phosphorylation of eIF2α is induced within 15 min of amino acid starvation (Zaborske et al., 2009). This means elevated concentrations of deacylated tRNA can occur even if only transiently before eIF2 phosphorylation dampens translation. We have rephrased the text to make this clearer.

Reviewer #2:1) The mechanistic implications of SERBP1•eEF2 complex remain unclear. Based on the finding of SERBP1 bound together with eEF2, the authors propose that SERBP1 depletes cellular eEF2. First, it would be helpful to expand the discussion and mention that yeast homologue Stm1 binds yeast ribosomes without eEF2. Do the mechanisms of translation regulation by Stm1 and SERBP1 differ in that the latter depletes not only the 80S but also eEF2?

The thinking that Stm1 binds the yeast ribosome in the absence of eEF2 is influenced by the absence of eEF2 in structures of the yeast 80S in complex with Stm1 (Ben-Shem et al., 2011). However, work from Nono Takeuchi-Tomita’s lab provides evidence that Stm1 stabilizes eEF2 on the yeast ribosome (Hayashi et al., 2017). It is therefore possible that Stm1 and SERBP1 function similarly within cells to deplete eEF2, but yeast eEF2 was lost during purification or did not crystallize.

Alternatively, the authors may have missed 80S•SERBP1 particles in their data set. Such particles may even be more abundant than 80S•SERBP1•eEF2 but also more dynamic, thus difficult to average and classify. Second, the mechanistic description of how SERBP1 prevents eEF2 dissociation remains vague. Comparison with other available eEF2 structures (with sordarin or other drugs and/or GDPNP/GDP nucleotides) might bring insights into how the conformational dynamics of eEF2 and/or the ribosome are affected by SERBP1.

We agree with the reviewer that SERBP1 probably binds ribosomes in the absence of eEF2 and cannot discount the presence of 80S•SERBP1 in our dataset. However, as eEF2 is one of the most highly expressed proteins in cells, ribosome•SERBP1 complexes may quickly recruit eEF2.

The exact mechanism for how SERBP1 prevents eEF2 from dissociating from the ribosome is difficult to infer from our two structural snapshots. However, it is clear that SERBP1 does not prevent GTP hydrolysis or induce large displacements in orientations of the domains of eEF2. Rather, its mechanism of action may be similar to sordarin, an antibiotic that also does not prevent GTP hydrolysis or induce large structural changes yet still traps eEF2 on the ribosome. Sordarin binds in a pocket between domains III, IV and V and in doing so increases interdomain contacts and subtly constrain the domains in conformations that prevent release (Pellegrino et al., 2018). As SERBP1 interacts only with domain IV of eEF2, it is likely that SERBP1 constrains the interdomain movements necessary for release by locking domain IV in a fixed position in the A site. We have altered the text to include a more detailed proposal for how we expect SERBP1 to prevent eEF2 dissociation.

2) IFRD2 is proposed to regulate translation by binding to non-translating 80S. But could IFRD2 also bind vacant 40S and prevent translation initiation?

IFRD2 interacts with both the small and large subunits, suggesting that the interaction is most stable with 80S ribosomes. Additionally, our immunoprecipitation experiments with FLAG-tagged IFRD2 did not isolate any 40S•IFRD2 complexes (as judged from analysis of our micrographs). Despite these observations, we cannot preclude the possibility that IFRD2 binds vacant 40S subunits before recruiting 60S subunits and in some cases a tRNA bound at the Z site. Regardless of the assembly pathway, IFRD2 would function to repress translation.

3) Describe the rotational state of the 80S bound with P-tRNA and Z-tRNA (likely non-rotated, but it would help to state this in the paper). Does IFRD2 induce a different 80S conformation?

The ribosome is in a non-rotated state in the presence of IFRD2 and/or Z-site tRNA. We have added this information to the manuscript.

Reviewer #3:[…] There is little information in the present manuscript that describes how the complexes examined here were specifically obtained from the reticulocyte lysate, but it would appear that they work under conditions in which only 80S particles form and that the isolated 80S peak is filled with a diverse array of distinct ribosome complexes. In the Brown et al., 2015 paper it appears to describe a FLAG-tag pull down of the nascent peptide, where focus was given to the eRF1-trapped sub-population of complexes. One of the 9 complexes defined here appears to be the same structure as that which was published in Brown et al., 2015b (the eRF1-ABCE1 POST complex), indicating a duplication or positive control of sorts. While I might be able to understand how the other P-site tRNA-containing complexes were isolated by a nascent polypeptide-directed pull-down approach, it is unclear how the complexes lacking P-site tRNA were obtained (i.e. the focus of the present investigations). I would strongly suggest further clarification on these points for the sake of reproducibility.

The original aim of the purification strategy was to isolate 80S•P•E•eRF1•ABCE1 complexes using a FLAG-tagged nascent chain and a catalytically inactive mutant of eRF1 in cell-free translational reactions. The complexes lacking peptidyl-tRNA are likely ribosomes that either co-associate with the target ribosome-nascent-chain complexes or bind non-specifically to the anti-FLAG beads during purification. This apparent co-association and/or non-specific binding also occurs in our experiments with FLAG-tagged IFRD2, where 75% of immunoprecipitated ribosomes observed by cryo-EM lack IFRD2. The phenomenon of achieving enrichment rather than homogeneity is widely reported with many target cryo-EM structures coming from about 10% of the total data, for example the ribosomal complexes described in Voorhees and Hegde, 2015; Shen et al., 2015; Braunger et al., 2018.

The two translationally inactive complexes discussed in detail in Figure 2, Figure 3 and Figure 4 are certainly interesting in regards to the mechanisms by which cells may "store" excess ribosomes and the quality of the data appear sufficient enough to support the claims that the authors make, which are largely qualitative descriptions of the SERBP1 and IFRD2 binding sites and the global positioning of the deacylated tRNA within the E site. In this regard, I think the paper is perfectly suitable for publication.What appears to need further clarification and elaboration is the discussion surrounding the "Z" site for tRNA binding. Alternative positions for E-site tRNA have been described since the early 80s in the early fluorescence studies of eukaryotic ribosomes (Robertson and Wintermeyer, 1987; Rodnina and Wintermeyer, 1992). Movements of deacylated tRNA within the E site have also been specifically described in recent single-molecule fluorescence studies (Ferguson et al., 2015 and references therein). Although the authors state that their "Z" site position may be similar or related to the position (Eout) described (Zhang et al., 2017), I don't see how they make this connection. I also don't understand the authors reference to the "known role [of the L1 stalk] in escorting deacylated tRNA from the ribosome", citing two bodies of work in bacteria that make no such claim or data to support such a claim. In this regard, is it really wise to give it a name other than E' or something of the sort? I fear that re-naming it the "Z" site will significantly complicate its description in future publications. The new position the authors see is clearly an "E site" position, yes?Aside from these major points, the paper certainly seems publishable in its present format from my perspective.

We decided to call the site as the “Z site” for a number of reasons. Firstly, the naming of E-site intermediates in bacteria is confusing and we did not want to add to this confusion. Published terms include Eout (Zhang et al., 2017), E2 (Agrawal et al., 2000), E´ (Robertson et al., 1986), and F site (Wower et al., 2000). We felt a different letter would help distinguish the site from reported intermediates in bacterial ribosomes. Secondly, using E´ (or similar) may give the impression that this tRNA position is definitely an intermediate of ejection from the E site. Although this is a plausible explanation, we cannot ignore the possibility that a tRNA can only bind to the Z site when the normal passage of tRNAs through the ribosome is stopped. Thirdly, “Z” as the last letter of the alphabet evokes the extreme position on the ribosome adopted by the tRNA.

The comparison with Eout (Zhang et al., 2017) was based on the way in which the anti-codon stem-loop of the tRNAs in both complexes interact with the extremity of the small subunit. However, given that the Z-site is specific to eukaryotes, we have removed this comparison. We have also removed the references to the role of the L1 in escorting deacylated tRNAs, except to say that the L1 stalk interacts with all known tRNA ejection intermediates.